# Vitrification of Pronuclear Zygotes Perturbs Porcine Zygotic Genome Activation

**DOI:** 10.3390/ani12050610

**Published:** 2022-02-28

**Authors:** Tengteng Xu, Chengxue Liu, Mengya Zhang, Xin Wang, Yelian Yan, Qiuchen Liu, Yangyang Ma, Tong Yu, Anucha Sathanawongs, Jun Jiao, Zubing Cao, Yunhai Zhang

**Affiliations:** 1Anhui Province Key Laboratory of Local Livestock and Poultry, Genetical Resource Conservation and Breeding, College of Animal Science and Technology, Anhui Agricultural University, Hefei 230036, China; xutengteng2022@163.com (T.X.); 15155138871@163.com (C.L.); zmy2021@stu.ahau.edu.cn (M.Z.); wangxin2022@stu.ahau.edu.cn (X.W.); yelian2021@stu.ahau.edu.cn (Y.Y.); qiuchenliu1998@stu.ahau.edu.cn (Q.L.); mayang890301@163.com (Y.M.); yt8504@ahau.edu.cn (T.Y.); 2Department of Veterinary Biosciences and Veterinary Public Health, Faculty of Veterinary Medicine, Chiang Mai University, Chiang Mai 50100, Thailand; anucha.sa@cmu.ac.th; 3Anhui Haoyu Animal Husbandry Co., Ltd., Luan 237451, China; jiaojun1678@163.com

**Keywords:** pig, vitrification, embryo, zygotic genome activation, blastocyst

## Abstract

**Simple Summary:**

Vitrification is a commonly used cryopreservation technique that has played an important role in the preservation of animal genetic resources and assisted reproduction in humans. However, the effect of vitrification on embryonic genome activation remains to be elucidated. This study found that vitrification reduced the developmental efficiency of pig blastocysts. Further studies found that vitrification reduced transcriptional activity during embryonic genome activation and disrupted gene expression. Our findings suggest that vitrification impairs genome activation in pig embryos, and our results can provide new insight into improving the developmental efficiency of vitrified embryos.

**Abstract:**

Zygotic genome activation (ZGA) plays an essential role in early embryonic development. Vitrification is a common assisted reproductive technology that frequently reduces the developmental competence of embryos. However, the effect of vitrification on porcine ZGA and gene expression during ZGA remains largely unclear. Here, we found that vitrification of pronuclear zygotes derived from parthenogenetic activation (PA) and in vitro fertilization (IVF) resulted in a significant reduction in the rates of 2-cell, 4-cell, and blastocysts, but did not affect the quality of blastocysts. Functional research revealed that RNA polymerase II Inhibitor (α-amanitin) treatment significantly reduced global transcriptional activity and developmental efficiency of both 4-cell and 8-cell embryos, implying an essential role of ZGA in porcine early embryonic development. Furthermore, vitrification did not affect the synthesis of nascent mRNA of 2-cell embryos, but significantly inhibited global transcriptional activity of both 4-cell and 8-cell embryos, suggesting an impaired effect of vitrification on porcine ZGA. Correspondingly, the single-cell analysis showed that vitrification caused the downregulation or upregulation expression of maternal genes in 4-cell embryos, also significantly decreased the expression of zygotic genes. Taken together, these results indicated that vitrification of pronuclear zygotes impairs porcine zygotic genome activation.

## 1. Introduction

Vitrification is widely used to preserve animal genetic resources in agriculture and human gametes and embryos in reproductive medicine [1,2]. Vitrification has been successfully applied to mammalian embryos from different species, such as humans [3,4], mice [5], cattle [6], sheep [7], and pigs [8,9]. However, the vitrification frequently posed cryoinjuries of embryos, such as cytoskeletal damage [10,11], mitochondrial dysfunction [12], DNA structure breakage, chromosome abnormalities [13], cell apoptosis [10,14]. In addition, vitrification also caused reactive oxidative species (ROS) accumulation, defective epigenetic modifications [15], abnormal expression of imprinted genes [5], and failure of blastocyst formation [6,16]. However, the effect of vitrification on zygotic genome activation (ZGA) during early embryonic development remains unclear.

Zygotes after fertilization have to undergo the maternal to zygotic transition (MZT) and embryo development should be regulated by newly synthesized zygotic products [17,18]. The developmental stage of ZGA varies among different species. It was reported that ZGA commonly occurs at the 2-cell stage in mice [19], 4–8 cell stage in pigs [20], 8–16 cell stage in cattle [21], sheep [22], and humans [23]. Accumulating evidence indicated that ZGA plays a critical role in early embryonic development. For example, mouse embryos, arrested at the 2-cell stage and displayed abnormal ZGA upon the exposure of α-amanitin, an RNA polymerase II inhibitor [24,25]. Therefore, ZGA is an essential prerequisite for early embryonic development [20].

ZGA is tightly regulated via degradation and translation of maternal transcripts and expression of zygotic genes, respectively. First, degradation of maternal transcripts mainly occurs during the maternal-to-zygotic transition. The failure of maternal mRNA degradation impaired ZGA during early embryonic development [26,27]. Second, the translation of maternal mRNAs also plays an important role in ZGA. For example, chromatin remodellerCHD1 mediates ZGA to promote lineage differentiation in mouse embryos [28]. Studies showed that loss of maternal GLIS1 led to the failure of ZGA, in turn causing developmental arrest at the 16-cell stage in mice [29]. Mice embryos with maternal BRG1 depletion failed to initiate ZGA and were arrested at the two-cell stage [30]. Third, the nascent mRNAs produced by embryos are involved in regulating ZGA and embryonic development. A previous study showed that depletion of transcription factor DUX prevented early embryonic development and caused defective ZGA [31]. Developmental pluripotency-associated 2 (Dppa2) and Dppa4 modulate the expression of the DUX family, which, in turn, affects ZGA and early embryonic development [32,33]. Numerous studies have shown that environmental stresses have adverse effects on the degradation of maternal factors and the expression of zygotic genes in embryos. Vitrification as a stressor frequently causes abnormal expression of maternal and zygotic genes [16,27,33,34]. It is worth noting that vitrification impaired early embryonic development by inducing apoptosis and the expression of stress-related genes [35,36,37]. However, the effect of vitrification on transcriptional activity during ZGA is yet to be known.

In the present study, porcine embryos were used to explore the effect of vitrification on ZGA, preimplantation embryo development, and the expression of maternal and zygotic genes. These findings may provide new insight into cryoinjuries of porcine vitrified embryos.

## 2. Materials and Methods

All reagents used in this study were purchased from Sigma (Sigma-Aldrich, St. Louis, MO, USA) unless otherwise stated. Animal experiments were conducted in accordance with the Institutional Animal Care and Use Committee (IACUC, Hefei, China) guidelines under currently approved protocols at Anhui Agricultural University. The ethics code is SYXK2016-007.

### 2.1. Experimental Design

#### 2.1.1. The Effect of Vitrification on the Development of Parthenogenetic Activation (PA) and In Vitro Fertilization (IVF) Embryos

For PA pronuclear zygotes, 204 zygotes were used for vitrification, and 180 zygotes were without any treatment as the control group. Biological replicates were performed Aa least three times in the PA embryo. For IVF pronuclear zygotes, 221 zygotes were used for vitrification, and 226 zygotes were without any treatment as the control group. Biological replicates were performed at least three times in IVF embryos. Zygotes were cultured to blastocyst in PZM-3 medium [38]. The blastocysts in the control group (89) and vitrification group (54) were detected CDX2 expression by immunofluorescence (IF).

#### 2.1.2. The Effect of RNA Polymerase II Inhibitor on the Development of Embryos

One gram of α-amanitin was dissolved in 1 mL DMSO as a storage solution, and 1 µL of storage solution was diluted to 25 µg/mL [39] working concentration with 39 µL porcine zygote medium containing 3 mg/mL of BSA (PZM-3). One microliter of DMSO was added to 39 µL PZM-3 as the control group. The control group (121 zygotes) and α-amanitin group (161 zygotes) were cultured to blastocyst in PZM-3 medium. Four times biological replicates were performed in this part. The control and α-amanitin treated 4-cell and 8-cell embryos were collected to performed 5-ethynyl uridine (EU) staining, and at least 30 embryos in each group were used to calculate the fluorescence intensity.

#### 2.1.3. The Effect of Vitrification on the Transcription Activity and Gene Expression during ZGA

The control, α-amanitin treated, and vitrified 2-cell, 4-cell, and 8-cell embryos were collected to performed EU staining. At least 30 embryos in each group were used to calculate the fluorescence intensity by Image J. To detect the gene expression during ZGA, the control, α-amanitin treated, and vitrified 4-cell embryos were collected to performed single-cell qPCR. At least 10 biological replicates were in each group.

### 2.2. In Vitro Maturation of Oocytes

The ovaries were taken from a local slaughterhouse, stored in a vacuum flask containing saline, and transported to the laboratory within two hours. Follicular fluid at 3–6 mm in diameter was extracted with a 10 mL syringe. After 10–15 min of follicular fluid precipitation, cumulus-oocyte complexes (COCs) with three layers were selected under a stereomicroscope. COCs were washed more than three times in vitro maturation medium (IVM) (TCM-199 supplemented with 5% FBS, 10% porcine follicular fluid, 10 IU/mL eCG, 5 IU/mL hCG, 100 ng/mL L-Cysteine, 10 ng/mL EGF, 0.23 ng/mL melatonin, 2.03 × 10^−5^ ng/mL LIF, 2 × 10^−5^ ng/mL IGF-1, 4 × 10^−5^ ng/mL FGF2, 100 U/mL penicillin, and 100 mg/mL streptomycin) [40]. Then, appropriately 80–100 COCs were transferred to a 4-well plate containing 400 μL IVM cultured for 40–44 h at 38.5 °C, 5% CO_2_, and saturated humidity. After maturation, cumulus cells were removed using 1 mg/mL hyaluronidase. Select the oocytes that extruded the first polar body for subsequent experiments.

### 2.3. Parthenogenetic Activation

The oocytes having extruded the first polar body were washed two times with an activation medium (0.3 M mannitol supplemented with 0.1 mM CaCl2, 0.1 mM MgCl2, and 0.01% polyvinyl alcohol) and stimulated with a single direct current pulse at 156 V for 80 µs by using a cell fusion instrument (CF-150B, BLS, Budapest, Hungary). The activated oocytes were incubated in a chemically assisted activation medium (PZM-3 plus 10 μg/mL cytochalasin B and 10 μg/mL cycloheximide) for 4 h. The embryos were then cultured in a PZM-3 medium [38] for seven days.

### 2.4. In Vitro Fertilization

One milliliter of fresh semen from two boars was mixed in 3 mL porcine in vitro fertilization medium (PIVF) (40 mL cell culture water containing 113 mM NaCl, 3 mM KCL, 2 mM Tris, 11 mM D-Glucose, 7.5 mM CaCl2 H_2_O, 5 mM Pyruvic acid sodium, 2 mg/mL BSA and 2 mM caffeine), take 1 mL of the mixture Centrifuged at 500× *g* for 20 min in 3 mL 60% Percoll. The sperm were then washed twice with PIVF and diluted to 1 × 10^6^ sperm/mL. Then, sperm co-incubated with oocytes at PIVF medium for 5 h. After fertilization, zygotes were cultured in PZM-3 at 38.5 °C, 5% CO_2_, and 95% air with saturated humidity. 

### 2.5. Vitrification of Pronuclear Zygotes

Porcine embryo vitrification was conducted based on protocols published previously [34]. At 16 h after PA/IVF, pronuclear zygotes were treated in an equilibration solution of 7.5% ethylene glycol and 7.5% DMSO in PBS without Ca^2+^ and Mg^2+^ for 5–10 min, followed by exposure to a vitrification solution of 15% (*v*/*v*) EG, 15% (*v*/*v*) DMSO and 1.0 M sucrose for 50–60 s. Five embryos were loaded on the tip of each Cryotop and immediately plunged into liquid nitrogen. For thawing, the Cryotop was removed from liquid nitrogen and dipped into a thawing solution containing 1.0 M sucrose for 1 min at 37 °C. Then, embryos were transferred to a diluent solution containing 0.5 M sucrose for 5 min and washed two times in a handing medium (TCM-199 supplemented with 20% FBS) for 3 min. After warming, surviving pronuclear zygotes were placed in a PZM-3 droplet which was equilibrated with mineral oil for at least 4 h, and cultured under the condition of 38.5 °C, 5% CO_2_ saturated humidity to observe development.

### 2.6. Single Embryo Quantitative PCR

The cDNA of each gene is amplified from 4-cell embryos using the Single Cell Sequence-Specific Amplification Kit (P621-01, Vazyme, Nanjing, China). At least ten 4-cell were collected from each group, and each 4-cell was a biological replicate. Then, the assembly of PCR was prepared in AceQ^®^ qPCR SYBR Green Master Mix (Q111-02/03, Vazyme, Nanjing, China) and was run on StepOne Plus (Applied Biosystems, Carlsbad, CA, USA). For gene expression analysis, Firstly, the CT value is derived, and the 2^−∆∆CT^ value is calculated from the CT value. Then, the 2^−∆∆CT^ value of the control group was then calibrated to 1, and the 2^−∆∆CT^ values of the other groups were calculated from this calibrated value. At least six biological replicates were used for gene expression analysis in each group.

### 2.7. Immunofluorescence Staining (IF) for Blastocyst

IF was performed as previously described [41]. To investigate the effect of vitrification on blastocyst quality, we used IF to detect the number of trophoblast-specific markers of CDX2 and nuclear in the blastocyst. Blastocysts were fixed in 4% paraformaldehyde solution for 24 h, permeabilized with 0.5% Triton X-100 for 30 min, and then blocked with 2% BSA for 1 h. Blastocysts were incubated in primary antibodies (mouse anti-CDX2 (MU392A-UC, Biogenex, Fremont, CA, USA)) overnight at 4 °C. Then, the embryos were incubated for 1 h in secondary antibodies (Alexa Fluor 488 anti-mouse IgG, Invitrogen, A11029, Carlsbad, CA, USA) without light. After washing, embryos were counterstained using Hoechst 33342 for 10 min and were then loaded onto glass slides. Finally, embryos were imaged using an inverted fluorescence microscope (IX71, Olympus, Japan). The trophoblast cell number and total cell number of blastocysts were determined using Image J. The number of inner cell mass is subtracted from the number of trophoblast cells by the total cell number.

### 2.8. EU Staining

The embryos were treated with 2 mM EU medium (PZM-3) for 1 h at 38.5 °C. The embryos were then placed in 4% PFA, fixed at room temperature (RT) for 15 min. Then, the embryos were placed in 0.5% Triton X-100, permeable for 20 min at RT. Next, place the embryos in the Click-iT^®^ reaction cocktail (C10329, Invitrogen, CA, USA) and incubate for 30 min at RT. The incubated embryos were then washed once with a Click-iT^®^ reaction rinse buffer for 5 min. Then, the embryos were placed in Hoechst 33342 incubated for 15 min at RT, washed three times with DPBS. Finally, samples were imaged at the same exposure value using an inverted fluorescence microscope (Olympus, Japan). For fluorescence intensity analysis, Image J was used to calculate the fluorescence intensity value of each embryo. The fluorescence image of the embryo was divided into different channels by Image J, and the fluorescence intensity of each embryo was obtained by subtracting the background fluorescence intensity from the total fluorescence intensity value. At least 30 embryos in each group were used to calculate the fluorescence intensity.

### 2.9. Statistical Analysis

All experiments were repeated more than three times. The data were analyzed by one-way ANOVA or *t*-test (SPSS 17.0). Differences of *p <* 0.05 were considered to be statistically significant. Differences in the mean percentages of 2-cell, 4-cell, and blastocysts among the groups were analyzed by *t*-test. The trophoblast (TE), inner cell mass (ICM) and total cell number of blastocysts, and the ratio of ICM to TE were also subjected to a *t*-test. The fluorescence intensity of embryos was analyzed by one-way ANOVA. The genes expression among each group also were analyzed by ANOVA.

## 3. Results

### 3.1. Vitrification of Pronuclear Zygotes Reduced the Developmental Efficiency of Parthenogenetic Activation Embryos 

To determine the effects of vitrification on developmental efficiency and quality of embryos, pronuclear zygotes were vitrified, and thawed embryos were cultured up to the blastocyst stage. Results showed that the rates of 2-cell (82.6%), 4-cell (59.4%), and blastocyst (25.8%) from frozen-thawed pronuclear embryos were significantly reduced compared to control group (97.4%, 86.6%, and 48.8%) (Figure 1A,B) (*p* < 0.05). To test the quality of blastocyst, total cell number and lineage allocation were analyzed in the resulting blastocysts. Blastocysts were stained with a CDX2 antibody to determine the total cell number and trophectoderm (TE) cells (Figure 1C). We found that vitrification of pronuclear zygotes did not affect the total cell number of blastocysts, TE, and ICM cell numbers (Figure 1D). Therefore, these results indicated that vitrification decreased the developmental competence of porcine embryos but did not affect the quality of blastocyst.

### 3.2. Vitrification of Pronuclear Zygotes Blocked Early Development of In Vitro Fertilization Embryos 

To further examine the effects of vitrification on the development of porcine early embryos, pronuclear zygotes derived from IVF were frozen and thawed embryos were cultured up to the blastocyst stage. Similarly, we observed that vitrification significantly reduced the rates of 2-cell (35.74%) and blastocyst (3%) compared to the control group (60.18% and 11.92%) (Figure 2A,B) (*p* < 0.05). These results showed that vitrification of pronuclear zygotes also decreased the early embryo developmental efficiency of IVF embryos.

### 3.3. RNA Polymerase II Inhibitor Inhibited the Early Embryo Development

To investigate the effect of vitrification on ZGA, we first analyzed the role of ZGA in the early development of porcine embryos. Pronuclear zygotes were exposed to RNA polymerase II inhibitor (α-amanitin) and cultured for seven days. EU staining analysis revealed that α-amanitin treatment significantly reduced the global transcriptional activity of both 4-cell and 8-cell embryos (Figure 3A) (*p* < 0.05). In addition, α-amanitin treatment significantly decreased the development efficiency of 2-cell (35.86%), 4-cell (24.73%), 8-cell (5.72%) and blastocysts (0%) compared to control group (94.98%, 63.56%, 48.06%, and 46.8%) (Figure 3B,C) (*p* < 0.05). Altogether, these results indicated that transcriptional activity during ZGA was essential for porcine early embryonic development.

### 3.4. Vitrification of Pronuclear Zygotes Impaired Porcine Zygotic Genome Activation 

To investigate whether vitrification affected porcine ZGA, EU staining was performed to detect the global transcriptional activity of embryos at the 2-cell, 4-cell, and 8-cell stages. Results showed that vitrification of pronuclear zygotes did not affect the global transcriptional activity of 2-cell embryos (Figure 4A,B). However, compared to the control group, vitrification significantly reduced the global transcriptional activity of 4-cell and 8-cell embryos (Figure 4C–F). Therefore, these results demonstrated that vitrification of pronuclear zygotes impaired porcine ZGA. 

### 3.5. Vitrification of Pronuclear Zygotes Perturbed Gene Expression during Zygotic Genome Activation 

To further elucidate how did vitrification affect ZGA in pigs, single-cell q-PCR was performed to examine the expression of maternal and zygotic genes during ZGA in 4-cell embryos. Meanwhile, α-amanitin was used to discriminate zygotic genes from maternal genes. Single qPCR revealed that vitrification did not change the expression levels of pluripotent genes including *CDX2*, *SOX2,* and *OCT4* (Figure 5A). However, vitrification resulted in a significant reduction in the expression levels of some zygotic genes, such as *GATA2*, *BMP4*, and *KRT8* (Figure 5B). Furthermore, vitrification significantly reduced the expression levels of some maternal genes, such as *HDAC8*, *KDM2B*, *TET2*, and *CDH2* (Figure 5C). Interestingly, vitrification also caused the increased expression of some maternal genes (Figure 5D). Together, these results indicated that vitrification of pronuclear zygotes perturbed gene expression during porcine ZGA.

## 4. Discussion

Vitrification of embryos is an important approach to protect endangered species and preserve the genetic resources of farm animals. At present, embryo vitrification has been successfully applied to humans [3], mice [5], cattle [6], sheep [7], pigs [8], but the developmental competence of vitrified embryos is still low. Studies showed that vitrification impaired embryonic development by inducing the defective cytoskeleton [11], oxidative stress, apoptosis [34,42], and abnormal epigenetic modifications [15]. However, the effects of vitrification on ZGA have not been reported. In this study, we found that vitrification of pronuclear embryos apparently reduced the blastocyst rate but did not affect the quality of blastocysts. In addition, porcine embryos uponα-amanitin treatment arrested at the 4-cell to 8-cell stage. Vitrification inhibited the transcriptional activity in 4-cell and 8-cell embryos. Therefore, our results indicate that vitrification inhibits the transcriptional activity of porcine embryos.

Previous studies indicated that vitrification impaired the developmental competence of embryos in humans [4], mice [43], pigs [34], and cattle [6]. Consistent with these studies, we found that vitrification significantly reduced the rates of cleavage and blastocysts. However, vitrification did not damage the quality of blastocysts. There are no significant differences in the total cell number of blastocysts and the number of trophectoderm cells. Similar results have been observed in other studies [15]. In contrast, some studies have shown that vitrification could disrupt the blastocyst quality [34]. This discrepancy could be due to the diversity of the species and the differences in embryo vitrification stages. Since gene expression in early embryos is sensitive to vitrification [33,44,45], the reduced developmental competence of vitrified pig embryos may be attributed to the abnormal gene expression during early embryo development.

It is well known that the degradation of maternal factors and the activation of the embryonic genome are critical to early embryonic development [18]. After fertilization, the embryonic genome is initially transcriptionally silent, and maternal transcripts and proteins regulate the development of early embryos. The genome is then activated through a process called maternal-to-zygotic transition, allowing the zygotic gene products to replace the maternal supplies that support the consequent development of embryos. In this process, some maternal and zygotic genes are critical for ZGA startup and early embryonic development in mammals. Studies showed that loss of maternal BRG1 [30], maternal GLIS1 [29], led to the failure of ZGA and blastocyst formation. It was reported that depletion of zygotic DUX induced defective ZGA and blocked early embryonic development [31,32,33]. Our results showed that vitrification impaired porcine ZGA and perturbed the expression of maternal and zygotic genes. 

Previous studies indicated that vitrification altered the expression of maternal factors important for early embryonic development. For example, maternal FGFR2 is important for the differentiation of trophectoderm and the establishment of cell polarity in early mouse embryos [46]. Sirtuin inhibition led to the developmental arrest of embryos and oxidative stress in both pigs and mice [47]. Moreover, vitrification could induce abnormal epigenetic modifications [15,48]. In this study, vitrification caused a significant reduction in the expression levels of key epigenetic genes, such as HDAC8, KDM2B, SMYD3, and TET2. Studies showed that histone methyltransferase SMYD3 regulates the expression of transcriptional factors during bovine early embryonic development [49]. Inhibition of TET protein activity in bovine embryos impaired DNA methylation reprogramming, reduced the efficiency of blastocyst development, and increased the number of apoptotic cells of blastocysts [50]. Therefore, the low developmental efficiency of vitrified pig embryos may be caused by the abnormal expression of these key epigenetic genes.

On the other hand, zygotic genes play an important role in early embryonic development. Studies indicated that chemokine ligand 24 (CCL24) regulates both the cell fate of ICM in bovine blastocysts and the first cell lineage differentiation [51]. The transcription factor GATA2 regulates cell polarity and affects early embryonic development [52]. In human preimplantation embryos, BMP4 knockdown or overexpression impeded blastocyst formation and triggered apoptosis in blastocysts [14]. In this study, we found that vitrification reduced the expression of genes important for embryonic development in porcine embryos. The potential mechanism by which vitrification alters gene expression is unclear. Vitrification could induce oxidative stress and apoptosis in embryos [5,34,53], and ZGA occurs during the maternal-to-zygotic transition. External stressors adversely affect the expression of maternal and zygotic genes during the maternal-to-zygotic transition. In this study, we found that vitrification of porcine pronuclear zygotes altered the expression of zygotic genes. It is thus possible that vitrification would perturb the expression of zygotic genes to impair ZGA in porcine embryos. However, the specific mechanisms underlying the altered expression of genes in vitrified porcine embryos still warrant further study.

## 5. Conclusions

In this study, these results indicated that. vitrification of pronuclear zygotes impaired porcine zygotic genome activation. Therefore, our findings would provide a theoretical basis for the optimization of vitrification procedures of porcine embryos.

## Figures and Tables

**Figure 1 animals-12-00610-f001:**
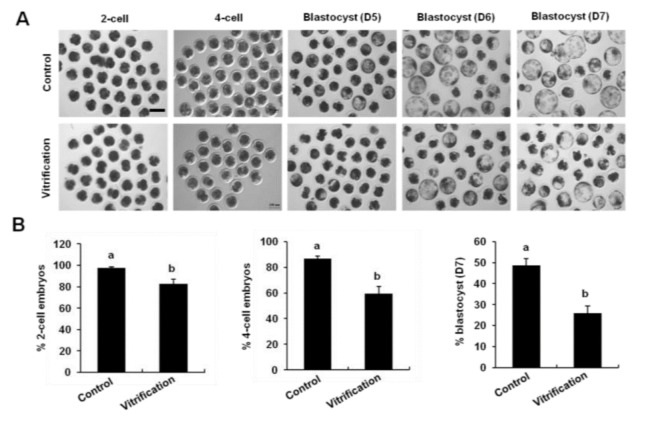
Effect of vitrification on the early development of parthenogenetically activated embryos and lineage allocation. (**A**) Pictures of embryos at different stages. Scale bar: 100 µm. (**B**) Developmental rates of embryos. The number of 2-cell (n = 175 and 168), 4-cell (n = 155 and 120), blastocyst (n = 89 and 54). (**C**) CDX2 (green) and DNA (red) staining of blastocysts. Scale bar: 50 µm. The number of the blastocyst (n = 89 and 54). (**D**) Cell lineage analysis of vitrification and control blastocysts. Total cell numbers, ICM cells, TE cells, and the ratio of ICM cells to TE cells were separately recorded and subjected to statistical analysis. Different markers represent significant differences. Five times biological replicates were used in this part.

**Figure 2 animals-12-00610-f002:**
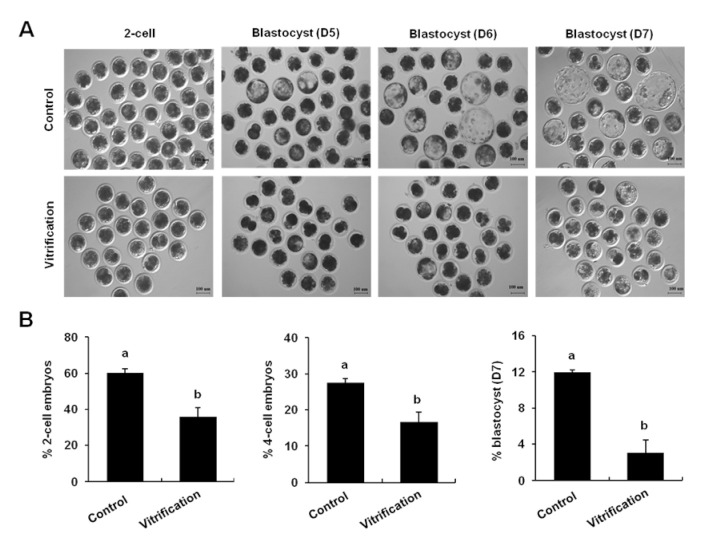
Effect of vitrification on the early development of in vitro fertilization embryos. (**A**) Pictures of embryos at different stages. Scale bar: 100 µm. (**B**) Developmental rates of pig IVF embryo. The number of 2-cell (n = 136 and 85), 4-cell (n = 62 and 35), blastocyst (n = 27 and 5). Different markers represent significant differences. Three times biological replicates were used in this part.

**Figure 3 animals-12-00610-f003:**
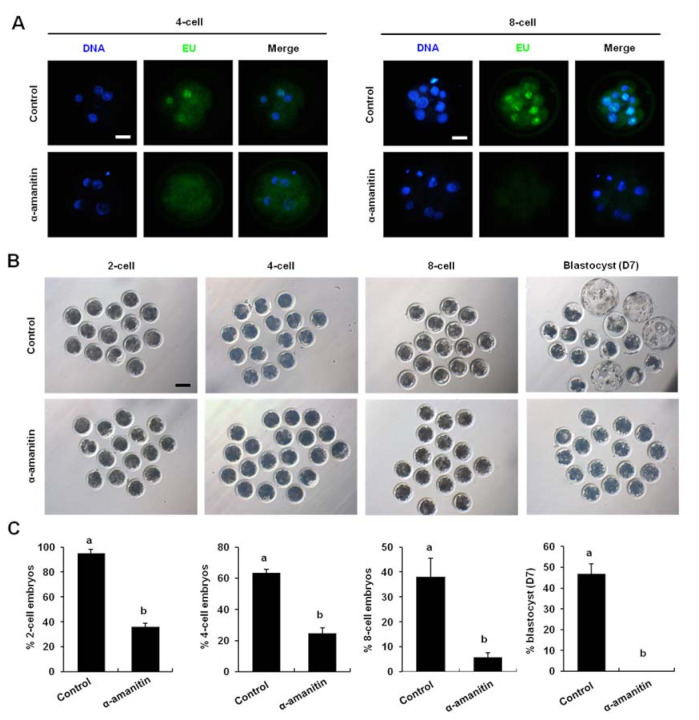
Effect of inhibition of transcriptional activity on parthenogenetically activated embryo development (**A**) α-amanitin can inhibit the transcriptional activity during the ZGA. The 4-cell and 8-cell were stained for EU (green) and DNA (blue). Scale bar: 100 µm. (**B**) Pictures of embryos at different stages. Scale bar: 100 µm. (**C**) Developmental rates of the porcine embryo. The number of 2-cell (n = 114 and 58), 4-cell (n = 76 and 40), 8-cell (n = 47 and 10), blastocyst (n = 40 and 0). Different markers represent significant differences. Four times biological replicates were used in this part.

**Figure 4 animals-12-00610-f004:**
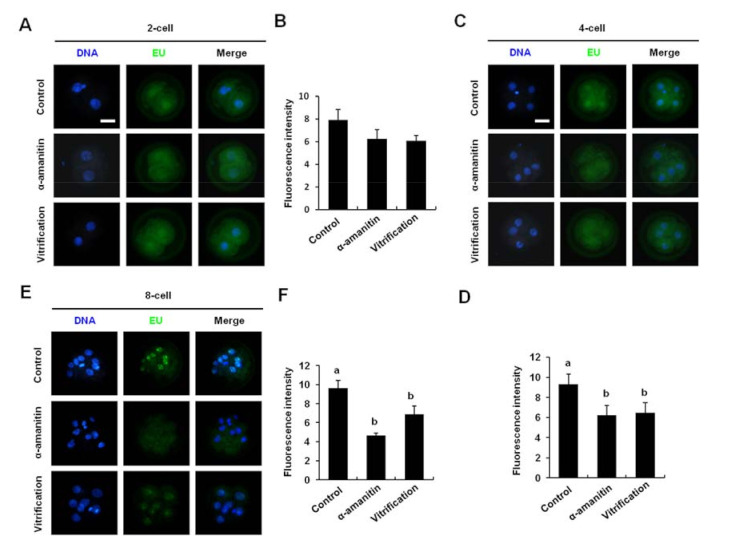
Effects of vitrification on porcine embryonic transcriptional activity (**A**,**B**) Effects of vitrification on transcriptional activity of 2-cell embryo. Scale bar: 100 µm. (**C**,**D**) Effects of vitrification on transcriptional activity of 4-cell embryo. Scale bar: 100 µm. (**E**,**F**) Effects of vitrification on transcriptional activity of 8-cell embryo. The 2-cell, 4-cell, and 8-cell were stained for EU (green) and DNA (blue). Scale bar: 100 µm. Fluorescence intensity was analyzed in three biological replicates with at least 20 embryos in each group. Different markers represent significant differences. Five times biological replicates were used in this part.

**Figure 5 animals-12-00610-f005:**
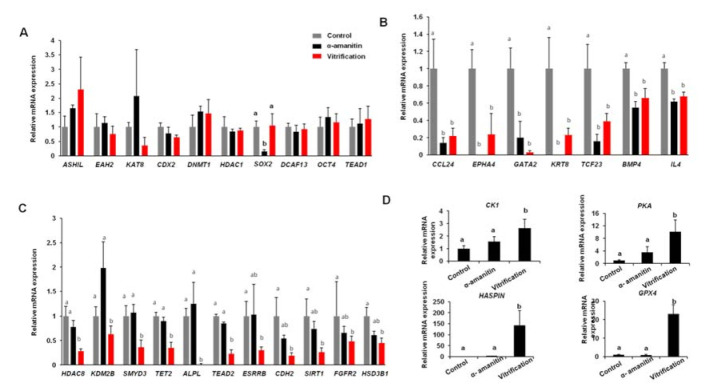
Vitrification alters gene expression in porcine 4-cell embryos (**A**) The expression of genes was unaffected by vitrification. The expression of the indicated genes in 4-cell was quantified by qPCR. (**B**) The expression of zygotic genes was down-regulated by vitrification. (**C**) The expression of maternal genes was down-regulated by vitrification. (**D**) The expression of maternal genes was upregulated by vitrification. At least six biological replicates were used for gene expression analysis in each group. Different markers represent significant differences. Eight times biological replicates were used in this part.

## Data Availability

The data presented in this study are available on request from the corresponding author.

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
