# Peer review of "Vitrification of Pronuclear Zygotes Perturbs Porcine Zygotic Genome Activation"

_animals, 2022, doi:10.3390/ani12050610_

Round 1
Reviewer 1 Report
The authors found that vitrification at the pronuclear stage reduced the proportions of PA and IVF embryos developing to the 2-cell, 4-cell, and blastocyst stages. The α-amanitin treatment also reduced the proportions of PA embryos developing to the 2-cell, 4-cell, and blastocyst stages. Vitrification at the zygote stage reduced transcriptional activity at the 4- and 8-cell stages, but not at the 2-cell stage. Vitrification at the zygote stage also reduced the relative transcript levels of several genes at the 4-cell stage.
The introduction provides suitable background information and the objectives of the study are clearly stated. The results are presented clearly. The discussion is concise and the conclusions are supported by the results. The authors should also clarify the potential mechanism (eg. stress) by which vitrification alters gene expression. This must be discussed in relation to the timing of post-activation events to compare and contrast the findings of other studies in vitrified oocytes and early embryos. I suggest the authors seek the assistance of a native English speaker to correct the numerous grammatical errors throughout the manuscript.
L82: Add “(PA)” after parthenogenetic activation to define the abbreviation.
L88: PA embryo replicates are already mentioned L85-86. Change “PA” to “IVF”.
L90, L97, L105, L109: Details of statistical analyses should be described in a separate sub-section of the Methods section.
L91: Define IF and CDX2.
L94: How was the concentration of α-amanitin chosen? Provide information of a dosage experiment, or include the relevant reference.
L96: “zygotes” appears twice.
L99: Define EU.
L111: How were the ovaries transported? In saline or PBS? At 38C? Please clarify.
L115: Give details of the IVM medium composition, including protein source (BSA?), growth factors (EGF?) and hormones.
L118: Provide the hyaluronidase concentration used.
L121: Give details of the activation medium composition.
L124: Define PZM-3 the first time it appears in the manuscript. Give details of the PZM-3 composition.
L128: Clarify whether the semen was freshly collected, extended, or frozen-thawed and give details of the semen preparation prior to gradient centrifugation.
L129: Give details of PIVF medium composition.
L132: Give the precise timing of vitrification for PA and IVF embryos relative to the start of activation and fertilisation procedures. Is there a reference for the vitrification method? What microdrop volume was used for the 5 embryos? What was the loading time? Give additional details and any relevant references.
L140: Give details of the handling medium composition.
Table 1: The table is not referred to in the text. The information in the table can be easily included in the text of the IF method (section 2.7).
L169: Clarify whether the Click-iT solutions were used according to the manufacturer’s instructions.
L180: The statistical analysis method for each data set should be specified in more detail in this section.
References: The authors lists are missing the second author surname. Please correct all the lists of authors.
Author Response
The authors found that vitrification at the pronuclear stage reduced the proportions of PA and IVF embryos developing to the 2-cell, 4-cell, and blastocyst stages. The α-amanitin treatment also reduced the proportions of PA embryos developing to the 2-cell, 4-cell, and blastocyst stages. Vitrification at the zygote stage reduced transcriptional activity at the 4- and 8-cell stages, but not at the 2-cell stage. Vitrification at the zygote stage also reduced the relative transcript levels of several genes at the 4-cell stage.
The introduction provides suitable background information and the objectives of the study are clearly stated. The results are presented clearly. The discussion is concise and the conclusions are supported by the results. The authors should also clarify the potential mechanism (eg. stress) by which vitrification alters gene expression. This must be discussed in relation to the timing of post-activation events to compare and contrast the findings of other studies in vitrified oocytes and early embryos. I suggest the authors seek the assistance of a native English speaker to correct the numerous grammatical errors throughout the manuscript.
L82: Add “(PA)” after parthenogenetic activation to define the abbreviation.
Response: Thanks for the good suggestion. We have added (PA) after parthenogenetic activation.
L88: PA embryo replicates are already mentioned L85-86. Change “PA” to “IVF”.
Response: Thank you for the good reminder. We have Changed “PA” to “IVF”.
L90, L97, L105, L109: Details of statistical analyses should be described in a separate sub-section of the Methods section.
Response: Thanks for the good suggestion. We have described the details of statistical analyses in a separate sub-section of the Methods section.
L91: Define IF and CDX2.
Response: Thank you for the good reminder. We have changed it.
L94: How was the concentration of α-amanitin chosen? Provide information of a dosage experiment, or include the relevant reference.
Response: Thank you for the good suggestion. We have added the relevant reference.
L96: “zygotes” appears twice.
Response: Thank you for the good reminder. We have deleted it.
L99: Define EU.
Response: Thank you for the good reminder. We have defined it.
L111: How were the ovaries transported? In saline or PBS? At 38C? Please clarify.
Response: Thank you for the good reminder. The ovaries were stored in a vacuum flask con-taining saline and transported to the laboratory within two hours.
L115: Give details of the IVM medium composition, including protein source (BSA?), growth factors (EGF?) and hormones.
Response: Thank you for the good suggestion. We have added it in material.
L118: Provide the hyaluronidase concentration used.
Response: Thank you for the good suggestion. We have added it in material.
L121: Give details of the activation medium composition.
Response: Thank you for the good suggestion. We have added it in material.
L124: Define PZM-3 the first time it appears in the manuscript. Give details of the PZM-3 composition.
Response: Thanks for good suggestion. Porcine PZM-3 medium is a fully qualified chemical medium, referring to an article published by Koji Yoshioka et al. 2002.
L128: Clarify whether the semen was freshly collected, extended, or frozen-thawed and give details of the semen preparation prior to gradient centrifugation.
Response: Thanks for good suggestion. The semen was freshly collected, 1 mL fresh semen from two boars was mixed in 3 mL porcine in vitro fertilization medium (PIVF) prior to gradient centrifugation.
L129: Give details of PIVF medium composition.
Response: Thank you for the good suggestion. We have added it in material.
L132: Give the precise timing of vitrification for PA and IVF embryos relative to the start of activation and fertilisation procedures. Is there a reference for the vitrification method? What microdrop volume was used for the 5 embryos? What was the loading time? Give additional details and any relevant references.
Response: Thank you for the good suggestion. At 16 hours after PA/IVF, pronuclear zygotes were treated with vitrification. We have added the relevant reference.
L140: Give details of the handling medium composition.
Response: Thank you for the good suggestion. We have added it in material.
Table 1: The table is not referred to in the text. The information in the table can be easily included in the text of the IF method (section 2.7).
Response: Thank you for the good suggestion. We have changed it.
L169: Clarify whether the Click-iT solutions were used according to the manufacturer’s instructions.
Response: Thank you for the good suggestion. We used the Click-iT solutions according to the manufacturer’s instructions.
L180: The statistical analysis method for each data set should be specified in more detail in this section.
Response: Thank you for the good suggestion. We would do it.
References: The authors lists are missing the second author surname. Please correct all the lists of authors.
Response: Thank you for the good suggestion.

Reviewer 2 Report
Manuscript animals-1539902 – “Vitrification of pronuclear zygotes perturbs porcine zygotic genome activation”
SUMMARY AND BROAD COMMENTS
The study analyses the effect of zygote vitrification on embryo development, zygotic genome activation/transcriptional activity and gene expression. It can contribute to understand why vitrification impairs embryonic development and, eventually, to address this issue and improve the outcomes of vitrified embryos. This could be useful for biobanking and animal conservation purposes.
The study design is appropriate. Methods are coherent with the aim, although they need to be explained more in depth. Results and conclusions are well explained.
Acronyms need to be explained the first time they appear (e.g., ZGA at Line 44, IF at Line 91, EU at Line 99. PIVF at Line 129).
Although biological material was derived from a slaughterhouse, an ethical statement is missing.
In my opinion, the study is suitable for publication after English and minor scientific revisions.
SPECIFIC COMMENTS
Simple summary is missing
Introduction
Line 37: please specify in what terms “Vitrification has less impact on the embryo than slow freezing”.
Lines 78-79: Please consider rephrasing the sentence. It is unclear.
M&M
Producer of reagents is missing.
Line 87-88: “Three times biological replicates were performed in PA embryo” disagrees with Lines 85-86 (“Five times biological replicates were 85 performed in PA embryo”). Please modify the sentence for IVF embryos.
Lines 90-91: Please add the reason why only PA embryos were used for Cdx2 staining (IVF embryos were not used).
Lines 93-95: Please double check the final concentration. With those dilutions, it should be 2,5 ug/uL.
Line 132: Please state how much time after PA/IVF was vitrification performed.
Line 141: Please explain how the post-warming survival of zygotes was evaluated.
Line 144: Please add information on nucleic acid extraction and bioinformatic analysis.
Line 178: Please add more details on how fluorescence intensity was evaluated (exposure, gain, focus, calculations).
Results
Line 262: OOCT2 is reported in the text, but OCT4 is reported in the related figure. Is that correct?
Figures
Figure 1A: scale bars are different in size in 2-cells and 4-cells pictures.
Figure 3A: scale bar size should be reported in the figure legend.
Figure 4: scale bar size should be reported in the figure legend.
Figure 4E: scale bar is missing.
It would be nicer to see if all the figures (1,2,3) had the same scale bar style.
Figure 5: It is unclear why 5D has a different style
Discussion/Conclusions
Line 337: “…the mechanism of vitrification is still not clear” might be misleading, especially in the conclusive paragraph.
Author Response
SUMMARY AND BROAD COMMENTS
The study analyses the effect of zygote vitrification on embryo development, zygotic genome activation/transcriptional activity and gene expression. It can contribute to understand why vitrification impairs embryonic development and, eventually, to address this issue and improve the outcomes of vitrified embryos. This could be useful for biobanking and animal conservation purposes.
The study design is appropriate. Methods are coherent with the aim, although they need to be explained more in depth. Results and conclusions are well explained.
Acronyms need to be explained the first time they appear (e.g., ZGA at Line 44, IF at Line 91, EU at Line 99. PIVF at Line 129).
Response: Thank you for the good reminder. We have amended it.
Although biological material was derived from a slaughterhouse, an ethical statement is missing.
Response: Thank you for the good reminder. We will add the ethic code SYXK2016-007.
In my opinion, the study is suitable for publication after English and minor scientific revisions.
Response: Thank you for the good suggestion.
SPECIFIC COMMENTS
Simple summary is missing
Response: Thank you for the good reminder. We have added the simple summary.
Introduction
Line 37: please specify in what terms “Vitrification has less impact on the embryo than slow freezing”.
Response: Thank you for the good suggestion. We have changed this sentence.
Lines 78-79: Please consider rephrasing the sentence. It is unclear.
Response: Thank you for the good suggestion. We have changed it.
M&M
Producer of reagents is missing.
Response: Thank you for the good suggestion. We have added producer of reagents.
Line 87-88: “Three times biological replicates were performed in PA embryo” disagrees with Lines 85-86 (“Five times biological replicates were 85 performed in PA embryo”). Please modify the sentence for IVF embryos.
Response: Thank you for the good suggestion. We have corrected this error.
Lines 90-91: Please add the reason why only PA embryos were used for Cdx2 staining (IVF embryos were not used).
Response: Thank you for the good suggestion. In this study, only 5 of the 221 vitrified frozen IVF embryos developed to blastocysts. We did not used IVF embryos to CDX2 staining due to the low number of blastocysts.
Lines 93-95: Please double check the final concentration. With those dilutions, it should be 2,5 ug/uL.
Response: Thank you for your attention. We have corrected this error. 1g α-amanitin was dissolved in 1mL DMSO as storage solution, and 1µL of storage solution was dilute to 25 µg/mL working concentration with 39 µL porcine zygote medium containing 3 mg/ml of BSA (PZM-3). 1 µL DMSO added to 39 µL PZM-3 as the control group.
Line 132: Please state how much time after PA/IVF was vitrification performed.
Response: Thank you for the good suggestion. We have added it.
Line 141: Please explain how the post-warming survival of zygotes was evaluated.
Response: Thank you for the good suggestion. We evaluated the survival rate after warming by observing whether the perivitelline space of the zygote was normal and whether the cytoplasm was uniform.
Line 144: Please add information on nucleic acid extraction and bioinformatic analysis.
Response: Thank you for the good suggestion. In this study, single-cell quantitative PCR was used to detect gene expression. The kit can directly reverse transcribe cDNA from embryos without the need for a separate nucleic acid extraction step.
Line 178: Please add more details on how fluorescence intensity was evaluated (exposure, gain, focus, calculations).
Response: Thank you for the good suggestion. We'll add more details about how fluorescence intensity was evaluated.
Results
Line 262: OOCT2 is reported in the text, but OCT4 is reported in the related figure. Is that correct?
Response: Thank you for your attention. We have corrected this error.
Figures
Figure 1A: scale bars are different in size in 2-cells and 4-cells pictures.
Response: Thank you for your attention. We have replaced these images.
Figure 3A: scale bar size should be reported in the figure legend.
Response: Thank you for the good suggestion. We will add the scale bar size in the figure legend.
Figure 4: scale bar size should be reported in the figure legend.
Response: Thank you for the good suggestion. We will add the scale bar size in the figure legend.
Figure 4E: scale bar is missing.
Response: Thank you for your attention. We will add the scale bar in this figure.
It would be nicer to see if all the figures (1,2,3) had the same scale bar style.
Response: Thank you for the good suggestion. We have replaced these images.
Figure 5: It is unclear why 5D has a different style
Response: Thank you for the good suggestion. Compared with the control group, the expression of HASPIN was very high after vitrification. The expression of these genes could not be clearly reflected if they were displayed in the same bar chart.
Discussion/Conclusions
Line 337: “…the mechanism of vitrification is still not clear” might be misleading, especially in the conclusive paragraph.
Response: Thank you for your attention. We have changed this sentence.

Round 2
Reviewer 1 Report
There are still numerous minor grammatical errors throughout the manuscript.
L213: Revise “total number” to “total cell number”. Also, the abbreviations for TE and ICM should be defined where the full terms are first used at lines 191 and 192.
L104: Add “[52]” after “PZM3” to cite the Yoshioka et al. 2002 reference.
References: Many of the author lists are missing the second author surname. Correct the formatting of the references and check that all author names are included correctly.
Author Response
Open Review
(x) I would not like to sign my review report
( ) I would like to sign my review report
English language and style
( ) Extensive editing of English language and style required
(x) Moderate English changes required
( ) English language and style are fine/minor spell check required
( ) I don't feel qualified to judge about the English language and style
Response: Thanks for the good suggestion. We have revised the English language.
Yes Can be improved Must be improved Not applicable
Does the introduction provide sufficient background and include all relevant references?
(x) ( ) ( ) ( )
Is the research design appropriate?
(x) ( ) ( ) ( )
Are the methods adequately described?
(x) ( ) ( ) ( )
Are the results clearly presented?
(x) ( ) ( ) ( )
Are the conclusions supported by the results?
(x) ( ) ( ) ( )
Comments and Suggestions for Authors
There are still numerous minor grammatical errors throughout the manuscript.
L213: Revise “total number” to “total cell number”. Also, the abbreviations for TE and ICM should be defined where the full terms are first used at lines 191 and 192.
Response: Thank you for the good reminder. We have changed it.
L104: Add “[52]” after “PZM3” to cite the Yoshioka et al. 2002 reference.
Response: Thank you for the good suggestion. We have added this reference.
References: Many of the author lists are missing the second author surname. Correct the formatting of the references and check that all author names are included correctly.
Response: Thank you for the good reminder. We have checked the format of the references and the author information.
